# Ultrastructural Changes of the Peri-Tumoral Collagen Fibers and Fibrils Array in Different Stages of Mammary Cancer Progression

**DOI:** 10.3390/cells14131037

**Published:** 2025-07-07

**Authors:** Marco Franchi, Valentina Masola, Maurizio Onisto, Leonardo Franchi, Sylvia Mangani, Vasiliki Zolota, Zoi Piperigkou, Nikos K. Karamanos

**Affiliations:** 1Department for Life Quality Study, University of Bologna, 47921 Rimini, Italy; 2Department of Biomedical Sciences, University of Padova, 35129 Padova, Italy; valentina.masola@unipd.it (V.M.); maurizio.onisto@unipd.it (M.O.); 3Department of Medicine, University of Bologna, 40100 Bologna, Italy; leonardo.franchi@studio.unibo.it; 4Biochemistry, Biochemical Analysis & Matrix Pathobiology Research Group, Laboratory of Biochemistry, Department of Chemistry, University of Patras, 26504 Patras, Greece; sylvia25.mangani@gmail.com (S.M.); zoipip@upatras.gr (Z.P.); 5Department of Pathology, School of Medicine, University of Patras, 26504 Patras, Greece; zol@med.upatras.gr

**Keywords:** breast cancer, collagen fibers, collagen fibrils, extracellular matrix

## Abstract

Breast cancer invasion and subsequent metastasis to distant tissues occur when cancer cells lose cell–cell contact, develop a migrating phenotype, and invade the basement membrane (BM) and the extracellular matrix (ECM) to penetrate blood and lymphatic vessels. The identification of the mechanisms which induce the development from a ductal carcinoma in situ (DCIS) to a minimally invasive breast carcinoma (MIBC) is an emerging area of research in understanding tumor invasion and metastatic potential. To investigate the progression from DCIS to MIBC, we analyzed peritumoral collagen architecture using correlative scanning electron microscopy (SEM) on histological sections from human biopsies. In DCIS, the peritumoral collagen organizes into concentric lamellae (‘circular fibers’) parallel to the ducts. Within each lamella, type I collagen fibrils align in parallel, while neighboring lamellae show orthogonal fiber orientation. The concentric lamellar arrangement of collagen may physically constrain cancer cell migration, explaining the lack of visible tumor cell invasion into the peritumoral ECM in DCIS. A lamellar dissociation or the development of small inter fiber gaps allowed isolated breast cancer cell invasion and exosomes infiltration in the DCIS microenvironment. The radially arranged fibers observed in the peri-tumoral microenvironment of MIBC biopsies develop from a bending of the circular fibers of DCIS and drive a collective cancer cell invasion associated with an intense immune cell infiltrate. Type I collagen fibrils represent the peri-tumoral nano-environment which can play a mechanical role in regulating the development from DCIS to MIBC. Collectively, it is plausible to suggest that the ECM effectors implicated in breast cancer progression released by the interplay between cancer, stromal, and/or immune cells, and degrading inter fiber/fibril hydrophilic ECM components of the peritumoral ECM, may serve as key players in promoting the dissociation of the concentric collagen lamellae.

## 1. Introduction

In 2023, breast cancer affected 2.26 million women and was the leading cause of cancer mortality among females [1]. The crucial pathogenetic event which can potentially lead to metastasis and patient death occurs when cancer cells invade the BM and ECM [2]. Understanding the mechanisms that drive the progression from DCIS to MIBC are crucial for understanding tumor invasion and preventing metastasis [3].

ECM drives the development of tissues and organs, plays vital functions, supports nutrients distribution, and acts as a biological barrier. It is a complex 3-dimensional (3D) scaffold which provides anchorage sites for cell adhesion and migration, favors or opposes cell movement, as well as connects but also separates different tissues [4,5,6,7,8]. Similarly, chemical signaling regulates a reciprocal interplay between cancer cells and the surrounding ECM microenvironment which can play a mechanical role in controlling tumor growth and driving cancer cell invasion [9,10,11,12,13,14,15,16,17,18,19,20,21,22,23]. The insoluble fibrillar collagen forming fibrils which bundle to form fibers, the hydrophilic molecules like glycosaminoglycans (GAGs) and proteoglycans (PGs) linking water, as well as fibronectin (FN) strongly contribute to the mechanical properties of ECM [24,25,26]. The fibrils exhibit hierarchical organization with alternating chirality at both molecular and supramolecular levels. This rope-like structural configuration provides enhanced resistance to tensile forces [27,28,29,30]. GAGs and PGs linking large amounts of water favor the mechanical sliding of fibrils/fibers and mainly oppose pressure loads [31,32].

Notably, collagen architecture directly regulates cell migration: densely bundled fibrils promote stable fibroblast adhesion, while pliable reticular fibrils facilitate adhesion retraction [33]. This structural dichotomy suggests that extracellular matrix organization serves as a biomechanical switch governing migratory behavior. Moreover, increased ECM stiffness represents a prognostically adverse microenvironment that promotes cancer cell invasion and metastasis. A denser tumor microenvironment including stiff fibrils of type I collagen promotes the growth of cytoplasmic protrusions and high-traction forces ability in cancer cells which assume migrating phenotypes [8,9,10,11,12,33,34,35]. In breast cancer both cancer and stromal cells increase the production of type I collagen fibrils, but with a different purpose. Breast cancer cells synthesize an aberrant homotrimeric type I collagen that assembles into thinner yet mechanically stiffer fibrils. These tumor-derived collagen structures exhibit dual pathological properties: (1) they actively promote cancer cell migration, and (2) are resistant to matrix metalloproteinase (MMP) degradation, a feature conserved across multiple solid tumor types [36,37,38,39].

On the other hand, the stromal cells surrounding the jamming ductal cells increase the deposition of normal heterotrimeric type I collagen in parallel and circular collagen fibers which limit cancer cell invasion both in vivo [13,40] and in vitro [16,17]. Tumor progression requires extensive remodeling of the peritumoral collagen architecture, which can biomechanically inhibit or promote cancer cell invasion [41,42]. Three different progressive stages have been described in a tumor-associated collagen signature classification (TACS): the first stage, or TACS-I, consists of a randomly arranged collagen fibril deposition (desmoplasia) around the breast cancer mass; in the second stage, or TACS-II, densely packed circular collagen fibers arranged parallel to the tumor surface constrain growth of the tumor mass and successfully oppose cancer cell invasion; in the third stage, or TACS-III, aligned and straight peri-tumoral stiff fibers follow a radial arrangement vs. the tumor surface to favor cancer cell invasion [12,16,43,44]. It was also suggested that straight and aligned stiff type I collagen fibers guide breast cancer cells in breaking the endothelial BM and entering the lymphatic and vascular systems (intravasation) [45].

The development from DCIS, corresponding to a TACS-II collagen array with no invading cancer cells, to a MIBC which is associated with the TACS-III stage and cancer invasion seems to be related to structural changes in the peri-tumoral ECM. In the TACS-II stage the densely packed peri-tumoral collagen fibers promote a local hypoxia which could induce the hypoxic cancer cells to stimulate stromal cells in secreting matrix proteolytic enzymes, involving MMPs, degrading the circular collagen fibers [46,47]. The dramatic evolution from TACS-II to TACS-III array has been mainly related to the recruitment of cancer-associated-fibroblasts (CAFs), and tumor-associated-macrophages (TAMs), which working together remodel the tumor microenvironment by secreting matrix proteolytic enzymes and depositing new radially aligned collagen fibers [19,48,49]. It was proposed that in invasive breast cancer, the aberrant expression of syndecan-1 in CAFs could induce the ECM collagen fiber alignment by interacting with FN-mediated fibrillogenesis via its heparan sulfate (HS) chains and by activating integrins favoring cell adhesion [50]. A tumor cell-secreted factor, matricellular protein WISP1, which binds to type I collagen may induce collagen fibers alignment or linearization. A cell-generated mechanical tension has been suggested to explain the alignment of the peri-tumoral collagen fibers, and a CAFs-generated tension theory has been proposed through an interplay between cancer cells and CAFs, which, working like myofibroblasts, could cooperate to modify the tumor ECM array and drive the linearization of the collagen fibers [51].

Therefore, with the purpose of evaluating the mechanism by which the circular collagen fibers of the TACS-II stage, successfully opposing cancer cell invasion in DCIS, are substituted by newly formed radial collagen fibers of TACS-III in MIBC, we carried out a correlative microscopy analysis to investigate, with the scanning electron microscope (SEM), the ultrastructural collagen architecture of fibers and fibrils in breast peri-tumoral ECM of human DCIS and MIBC deparaffinized histological sections.

## 2. Materials and Methods

### 2.1. Clinical Samples and Histology

Histological sections of biopsies from breast cancer patients (Table 1) were used to obtain tissue sections, ~4 μm thick, which were deparaffinized in xylene for 10–15 min and dehydrated using serial percentages of ethanol. To visualize and characterize the organization of collagen fibers, particularly types I and III, in tissue sections, these were stained with 5% (*v*/*v*) Picrosirius Red and observed under a Leitz Ortholux 2 polarized light microscope. Images were directly acquired in digital format with a Nikon DS-5M microscope camera.

### 2.2. Hematoxylin and Eosin Staining

Sections were deparaffinized and hydrated to distilled water. Adequate hematoxylin was applied to completely cover the tissue section, and sections were incubated for 5 min in RT. The slides were rinsed in two changes of distilled water (15 s each) to remove excess hematoxylin stains. Adequate Bluing reagent was applied to completely cover the tissue section and was followed by incubation for 10–15 s. The slides were rinsed again in two changes of distilled water (15 s each) to remove excess stains. The slides were dipped in 100% (*v*/*v*) ethanol for 10 s and the excess was blotted off. Adequate Eosin Y solution was applied to completely cover the tissue sections and was followed by incubation for 2–3 min. The slides were rinsed in 100% (*v*/*v*) ethanol for 10 s. Finally, the slides were dehydrated in three changes of 100% (*v*/*v*) ethanol (1–2 min each), the excess was blotted off, and the slides were mounted with a mounting medium before a coverslip was applied. The slides were observed under a Leitz Ortholux 2 polarized light microscope. Images were directly acquired in digital format with a Nikon DS-5M microscope camera.

### 2.3. Scanning Electron Microscopy

After the histological observations, the seriated sections of the same biopsies were immersed in xylene for 10–15 min at room temperature to remove paraffin. Fragments of glass supporting the deparaffinized tissue sections were dehydrated with ascending grades of alcohol and subsequently subjected to critical point drying. The specimens were mounted and immobilized on appropriate stubs by a conductive bi-adhesive tape and then coated with a 5 nm palladium gold film by a sputter-coater (Emitech 550 sputter-coater) to be observed under a SEM (Philips 515, Eindhoven, The Netherlands) operating in secondary-electron mode. Micrograph acquisition was obtained by a Pentax K-5 camera.

## 3. Results

### 3.1. Ultrastructural Analysis of Collagen Fibers and Fibrils Array in Ductal Carcinoma In Situ

Histological sections from DCIS showed polygonal ductal cancer cells confined in the ducts which were surrounded by buckled collagen fibers. Considering the TACS, this stage corresponded to the TACS-II stage which is characterized by densely packed and circular collagen fibers running parallel, close to the duct surface, acting like a valid barrier to oppose cancer invasion. A few immune cells were visible among the circular collagen fibers, mainly far from the ducts (Figure 1a). The ultrastructural analysis of the same deparaffinized histological sections showed that the circular collagen fibers surrounding the ducts intimately adhered to the cancer cells and actually looked like dense collagen fibers with no inter-fiber gap or nano space between the fibrils or next to the cancer cell BM. In these areas no cancer cell invaded the ECM, therefore confirming the mechanical role of the densely bundled fibers surrounding the ducts (Figure 1b). However, in other areas of the same sample 5 × 12 µm fusiform spaces, presumably containing water and resulting from a dissociation of the collagen fibrils, were detectable inside the circular collagen fibers, under the light microscope and SEM (Figure 1c,d).

A deeper ultrastructural analysis of the same sections demonstrated that the circular collagen fibers next to the ducts were comparable to concentric collagen lamellae composed of collagen fibers whose densely packed fibrils run longitudinally, obliquely or circularly vs. the duct axis. Collagen fibrils followed the same direction and run parallel to each other inside each lamella, whereas they coursed orthogonally vs. the adjacent lamellae (Figure 2a–d). A similar structural array seems to better confine the proliferating ductal cancer cells. In some regions the collagen-concentric orthogonal lamellae appeared dissociated and showed inter-lamellar spaces of less than 10 µm (Figure 2b–d). Despite these spaces probably containing water linked to PGs/GAGs, when the collagen fibers of the lamella next to the ducts appeared densely bundled they acted as a biological barrier: no cancer cell invasion, as well as extravesicle infiltration of the ECM, were visible both at a microscopic and at an ultrastructural level (Figure 3a,b). In some patients the concentric collagen lamellae seemed to substitute a primary desmoplastic peri-tumoral collagen meshwork, probably corresponding to TACS-I array and located next to the duct BM (Figure 3a). By SEM analysis, the fibrils composing the dense collagen fibril meshwork appeared as large collagen fibrils, ranging 100–120 nm in diameter (Figure 3b). The role of the mechanical pressure of the growing tumor in cancer cell invasion was confirmed by a jamming state inside some ducts, which induced cancer cells lying on the BM to partially protrude into the peri-tumoral ECM (Figure 3c). Protruding cancer cells confined by collagen fibers were confirmed with SEM by collagen fiber sheaths developing from the concentric collagen lamellae and surrounding the cancer cells lying on the BM (Figure 3d).

Histological sections of the other biopsy stained with Picro Sirius Red to evidence the course of collagen fibers under the Polarized Light Microscope showed inter-lamellar spaces between crimped collagen lamellae (Figure 4a). By SEM investigation these inter-lamellar gaps looked like regular fusiform spaces (10–15 µm wide) probably consequent to a dissociation of the concentric lamellae and corresponding to longitudinal channels containing water linked to PGs/GAGs (Figure 4b). The same deparaffinized histological sections by SEM investigations showed that the cancer cells inside the ducts produced many microvesicles on their cytoplasmic surface but were successfully confined by the intact concentric collagen lamellae (Figure 4c). At the light polarized microscope, when the polygonal proliferating jamming cancer cells filled the ducts, a partial disruption or dissociation of the concentric collagen lamellae was sometimes detectable (Figure 4d). By SEM analysis the regions of DCIS biopsies with a TACS-II collagen array showed a structural interruption of the concentric collagen lamellae which allowed an infiltration of exosomes and microvesicles in the peri-tumoral ECM (Figure 4e). In other areas of the same DCIS biopsy, the collagen fibers comprising the concentric orthogonal lamellae clearly changed their direction and were bent, even at 90°, to assume a radial array vs. the surface of the duct and corresponding to a dramatic TACS-III stage. Moreover, the inter-fiber spaces created by the radial collagen fiber array allowed the invasion of extravesicles which were visible only by SEM (Figure 4f).

### 3.2. Ultrastructural Analysis of Collagen Fibers and Fibrils Array in MIBC

In histological sections of human MIBC, some ducts were still surrounded by the CL described in DCIS, which however appeared clearly dissociated and separated by wide inter-lamellar spaces containing both invading cancer cells and a rich immune cell infiltration. In general, a collective cancer cell invasion of the ECM was always evident if a dissociation of the CL occurred. The grouped cell invasion into the inter-fiber spaces seemed to be a consequence of the transition from a jamming to an unjammimg state of the highly proliferating ductal cancer cells. Each collagen lamella appeared histologically intact and both cancer cells and immune ones seemed to be confined inside each inter-lamellar gap, even though the immune cells were more concentrated in the inter-fiber spaces of the external layers. This suggested the mechanical role that the CL dissociation could play in favoring cancer cell invasion. At high magnifications some grouped cancer cells invading the ECM seemed to be individually enveloped by dense eosinophilic material, presumably corresponding to opposing densely packed collagen fibers (Figure 5a). In the peri-tumoral ECM of the deparaffinized histological sections investigated by SEM, intercellular cytoplasmic protrusions, similar to tunneling nanotubes supporting direct intercellular communications between adjacent rounded immune cells, but also connecting these cells to the invading cancer cells, were detectable next to the duct (Figure 5b).

Picro Sirius Red-stained histological sections observed under the Polarized Light Microscope showed that the invading cancer cells in the peri-tumoral microenvironment were always in direct contact with or also surrounded by collagen fibers (Figure 5c). The same sections deparaffinized and then evaluated by SEM confirmed that collagen-like sheaths intimately surrounded the single invading rounded cancer cells which occasionally showed invadopodia developing from their ventral surface (Figure 5d). The development of cytoplasmic protrusions as well an intimate physical contact between cancer cells and collagen fibers suggested the crucial role of collagen in favoring the cancer cell migration.

In other MIBC biopsies corresponding to a TACS-III stage, the straight and radially arranged collagen fibers appeared in continuity with and deriving from the CL surrounding the ducts. Cancer cell invasion occurred at the inter-fiber spaces of the dissociated concentric collagen lamellae, as well as between the radially aligned collagen ones (Figure 6a–c). SEM analysis of the same sections confirmed that the CL running parallel to the duct surface changed their direction, assuming a radial array and delimiting inter-fiber channels used by invading cancer cells to further penetrate into the microenvironment (Figure 6d). These images suggested to us that the radially arranged collagen fiber around the ducts and described in TACS-III stage could develop by a directional change and wide dissociation of the circular, oblique, and longitudinal collagen fibers comprising the CL (Figure 6e). At the scanning electron microscope, the invading cancer cells penetrating the inter-fiber channels delimited by straight fibrils of radially arranged swollen fibers developed filopodia/invadopodia looking for collagen, thus demonstrating the physical interaction between cancer cell migration and collagen scaffold (Figure 6f).

In areas of advanced breast cancer MIBC, where collagen fibers forming the densely packed CL strongly adhering to the duct BM persisted as a biological barrier, no cancer cells invaded the tumor microenvironment. However, far from the ducts, collagen fibers following random directions and looking like radially aligned fibers were invaded by clusters of ductal cancer cells mixed with a conspicuous immune cell infiltrate (Figure 7a). These morphological images further suggest a crucial role of inter-lamellar spaces in favoring cancer cell invasion. In these areas, the Picrosirius Red stained sections observed under the Polarized Light Microscope revealed relatively thin, straight, and parallel fragmented fibers which act as a rail for cancer cell invasion as well as immune cell migration (Figure 7b). SEM observations of the deparaffinized histological sections confirmed the massive penetration of small, rounded immune cells which might contribute to creating inter-lamellar spaces in the peri-tumoral ECM (Figure 7c). The deparaffinized sections analyzed by SEM showed that invading cancer cells were driven by the parallel, straight, and thin collagen fibers acting like physical rails (Figure 7b,d–f). The polygonal or rounded invading cancer cells, recognizable for the larger size and the rough cytoplasmic surface including extravesicles, demonstrated to be looking for collagen by developing short filopodia and invadopodia or strongly adhering to the collagen fibers (Figure 7d,e). Far from the ducts, straight collagen fibers were particularly densely arranged, thick, and presumably stiffer so that no single fibrils were detectable. These fibers formed very straight inter-fiber channels (about 10 µm wide) in which single or grouped cancer cells producing and also releasing microvesicles and exosomes freely migrated (Figure 7f,g). In the same track or in different parallel channels single immune cells were also observed (Figure 7h). In particular, the immune cells which probably corresponded to T-lymphocytes or macrophages showed they were able to degrade and invade the ECM or move with an ameboid movement into inter-fiber channels by means of cytoplasmic protrusions (Figure 7i,j).

## 4. Discussion

In this study, histological sections from human biopsies of DCIS and MIBC are analyzed by SEM to investigate the role of the peri-tumoral collagen array in regulating cancer cell invasion. The 3D ultrastructural investigation demonstrated that the previously described “circular collagen fibers” in the TACS-II stage actually resemble circular concentric collagen lamellae (Figure 8). Collagen lamellae are composed of collagen fibers which include densely packed fibrils running in parallel inside each lamella but arranging the orthogonal vs. the adjacent ones. Such a fibrotic organization can better oppose the jamming DCIS cells and is very similar to biophysical barriers. Examples include the collagen lamellae in the annulus fibrosus of the intervertebral disks which restrain movements of the nucleus pulposus [53], in the osteon unit of the compact bone to resist gravity loads [54], in normal cornea [55] as well as in the circular ligament surrounding dental implants to hinder virus and bacteria penetration [56].

In DCIS showing the TACS-II stage, a growing pressure generated by the tumor mass expansion was confirmed by SEM investigation by thin sheaths of collagen fibers of the circular lamellae partially surrounding the protruding cancer cells which have been demonstrated to deform collagen fibers and interact with breast tumor microenvironment architecture (Figure 8) [57]. The peri-ductal collagen fibrils and fibers, which usually act like ropes and resist tension, appeared laterally compressed by ductal jamming pressure and therefore submitted to stretching and aligned in parallel [58,59]. The fibril/fiber tension induces a tissue structural shrinkage which makes the concentric collagen lamellae more compact, with no inter-fiber spaces, and mechanically resistant to cancer cell invasion. Moreover, SEM observations showed that cancer cells invading the ECM develop tight contacts with collagen fibrils which comprise the main component of the peri-tumoral nano-environment. Therefore, in addition to the architecture of the peri-tumoral collagen fibers, as well as the physical features of collagen fibrils, like an increase in fibril diameter and lysyl oxidase-mediated crosslinking observed in advanced stages of breast cancer, could regulate and drive the invasion of cancer cells into the ECM [60,61,62]. These considerations suggest that it might be crucial to investigate the ultrastructural changes in collagen arrangement among DCIS and MIBC. Although a statistical analysis of fibrils diameter in collagen lamelae by SEM analysis was not possible because the fibrils were very densely packed, single fibrils were discernable as they exhibited a large diameter (ca. 100–120 nm) which was compatible with the thick type I collagen fibrils favoring invasion, intravasation, and metastasis [63,64,65,66]. It has been demonstrated that type III collagen, the second fibrillar collagen in dermis, limits metastasis by sustaining tumor dormancy [67,68]. The different behavior of cancer cell vs. type I and III collagens is related to the fibril physical properties like diameter and flexibility. Stiff type I collagen fibrils, resisting high tensile strength like in the tendon, are tightly packed in straight fibers, show a large diameter, and are poorly elastic. Differently, type III collagen usually forms thinner and more elastic fibrils composing loose and very pliable fibers [65,69,70,71,72]. Fibril bending stiffness, independently of fibril thickness and intrafibrillar crosslinking, can favor epithelial-to-mesenchymal transition (EMT) of invasive and non-invasive breast cancer cells [61]. Indeed, bending stiffness of the collagen fibril has been related to the supramolecular helical array of microfibrils forming the single fibril: fibrils showing an almost parallel microfibril arrangement (5° angle) are larger and stiffer, whereas those fibrils displaying a more evident helical microfibril array (17° angle) are thinner and more flexible [71]. The dissociation of the peri-tumoral collagen lamellae create tracks or channels whose rigid walls can better favor cancer cell adhesion and drive cancer cell invasion (TACS-III) vs. the pliable and thinner type III collagen fibrils. To this end, the data obtained in this study clearly demonstrate that the mechanical properties of single fibrils can also drive cancer cell invasion in the tumor microenvironment (Figure 8).

It is worth noticing that DCIS biopsies that display a TACS-II stage showed a partial dissociation of the peri-tumoral collagen lamellae with fusiform gaps (10–15 µm wide). This is associated with immune cell recruitment and the movement of single invading cancer cells as well as extracellular vesicles (microvesicles and exosomes) which can transmit biological messages to CAFs favoring the TACS-III array [16,48,73,74,75]. It is therefore clearly demonstrated that the histological evaluations, which imply limits of resolution in evidencing a single cell or even more the sparse small extravesicles, should be supported by a deep ultrastructural analysis of the deparaffinized sections.

Another crucial point demonstrated in the present study is that even though the development from a DCIS to a MIBC stage has been associated with a rearrangement of the collagen fibers from a TACS-II to a TACS-III array, we observed biopsies of MIBC still showing the TACS-II stage. Notably, this is characterized by a dissociation of the collagen lamellae next to the ducts always associated with an intense infiltrate of immune cells [76,77,78]. The observed immune cells able to develop mutual intercellular cytoplasmic connections like tunneling nanotubes, but also with cancer cells, can be identified as tumor-associated macrophages (TAMs) [17,79].

In DCIS with TACS-II, the progressive tumor growth and cellular crowding induce an alternation of cell compression and stretch which promote cell jamming, increase in the peri-tumoral interstitial fluid pressure, cancer EMT invasion and CAFs activation for generating the newly formed radial aligned fibers in TACS-III [80,81,82,83,84]. However, our ultrastructural observation demonstrated that the peri-tumoral radially aligned tensioned fibers described in MIBC with TACS-III clearly develop from a mechanical bending of the same concentric orthogonal lamellae observed in the TACS-II stage. In MIBC biopsies, inter-fiber channels (10–15 µm wide) far from the ducts and delimited by very densely packed fibrils contained a few cancer cells developing filopodia and invadopodia or/and immune cells. Isolated extravesicles, whose epithelial or stromal origins were not discernable, were also present in narrower channels. The inter-fiber channels as well as early partial lamellar dissociation could be related to both mechanical pressure and enzymatic activity of both cancers and immune cells. If DCIS biopsies showed relatively poor immune cell infiltration, the mammary MIBC biopsies with cancer cell invasion displayed many immune cells filling and invading wide inter-fiber gaps. The dense concentric collagen lamellae surrounding the ducts in TACS-II oppose cancer cell invasion but also hinder the migration of tumor-infiltrating T cells [85]. When leukocytes need to move, they have to cross barriers like the BM and ECM by enzymatic activity involving MMPs and heparanase (HPSE) [19,86,87,88,89]. Particularly, cytotoxic T-lymphocytes migrating through the collagen network may create channels which will be available for other T-lymphocytes, but could contribute to an ECM dissociation [90]. At the same time, leukocyte-derived HPSE can enhance tumor immune surveillance, but they are also able to favor cancer cell invasion and tumor progression in tumors whose cells are also able to modulate the HPSE expression in lymphocytes [91].

PGs and GAGs linking collagen of the peri-tumoral ECM might regulate the array of the collagen lamellae around the mammary tumor [92]. The content of decorin, small leucine-rich proteoglycans (SLRPs) that specifically bind to collagen fibrils, changes from the TACS-II to the TACS-III stage in breast cancer. Decorin also suppresses bone metastasis in MDA-MB-231 breast cancer cells, and contributes to oppose cancer cell invasion as its stromal expression in breast invasive ductal carcinoma (IDC) is significantly weaker if compared to that in DCIS [93,94,95]. It binds to type I and II collagen, limits fibril diameter in fibrillogenesis and regulates the biomechanical properties of connective tissues [26,92,96]. Decorin could provide a mechanical strengthening of the fibrils inside each fiber of the concentric collagen lamellae surrounding breast cancer ducts [59,97,98]. Therefore, the lack of decorin favors ECM disruption, the swelling of fibrils as well as the development of stiff very large-diameter fibrils which favor adhesion and traction of the invading cancer cells [99,100].

In normal conditions, the hydrophilic GAGs and PGs, linking water, play a chemo-mechanical role in lubricating collagen fibrils and fibers to reduce friction, and contribute to regulating a filter which controls the migration of cells, virus, bacteria, and nutrients. GAGs-water aggregates can mechanically oppose compression. GAGs which link very large amounts of water could provide a biomechanical resistance to tumor growth, or also favor the interstitial flow and cancer cell invasion [92]. It is plausible to suggest that all the empty inter-fiber spaces analyzed by SEM in human DCIS and MIBC contain large amounts of water linked to ECM components, because both histological and SEM processing procedures require complete dehydration. In breast cancer, tumor stiffness induces an increase in the extracellular pressure which promotes HSPG formation and its protein syndecan-1 can align collagen into stiffer and parallel collagen fibers [101]. Moreover, the secretion of HPSE, an endoglycosidase enzyme produced by both cancer and immune cells, opens straight “highways” for cancer cell invasion and then metastasis by degrading the inter-fiber molecules of HS [19].

Water modulates ECM viscoelasticity and PGs regulate different proportions of loosely to strongly bound water, swelling, water binding or osmotic environments in connective tissues [102]. Cancer cells are able to change their mechanism of migration to ensure successful dissemination and metastasis. It is worth noticing that in our biopsies, we did not observe the elongated mesenchymal invading cancer cells [103,104]. The mechanical role of water in regulating MDA-MB-231 breast cancer cells migration was demonstrated by Stroka et al. [105]. The direction of breast cancer cell migration reversed in hypotonically shocked cells at the leading edges or hypertonically shocked cells at the trailing edges [105]. Notably, a hypertonic inter-fiber microenvironment activating a cytoplasmic water permeation in cancer cells could activate this mechanism of migration/invasion, in addition to the ameboid and mesenchymal ones. In breast cancer, a hypertonic microenvironment could be favored by a high concentration of hylauronan (HA), which is the most distributed PG in dermis, frequently upregulated in many cancers, and is associated with increased aggressiveness and poor prognosis in breast cancer [106,107,108]. Besides the important chemical interaction among cancer cells and signaling molecules, the hydrophilic HA reduces friction and favors cancer cell penetration [109]. Breast cancer cells release HA in the ECM by shedding HA-coated cytoplasmic microvesicles and exosomes from the tip of short cytoplasmic filopodia or microvilli [73,110,111]. In the present study, extravesicles with morphological characteristics of microvesicles and exosomes were observed both in DCIS and MIBC, dispersed in collagen fibers of circular and radial collagen lamellae (Figure 8).

Our data suggest that PGs and GAGs of the peri-tumoral ECM could play chemo-mechanical roles in regulating the structural architecture of peri-tumoral collagen lamellae array and cancer cell invasion in human ductal breast cancer. The above-mentioned pathways are summarized in Figure 8. The analysis of deparaffinized histological sections by SEM analysis may therefore be of significant additive value for a simple, fast, and correlative technique to obtain new data for clinical treatment and prognosis as well as to a full understanding of the morphological changes in the peri-tumoral ECM.

## 5. Conclusions

Changes in the ECM, particularly collagen array, contribute to breast cancer progression from DCIS to MIBC. Notably, the key findings of this study reveal that breast cancer progression is marked by profound ultrastructural changes in the peri-tumoral collagen architecture. Using SEM analysis on human biopsy samples, we found that in DCIS, collagen fibers form concentric, densely packed, lamellar structures that likely act as a physical barrier to cancer cell invasion. In contrast, as the tumor advances during the transition to MIBC, paracrine signaling activates fibroblasts and immune cells, leading to enzymatic remodeling, increased matrix stiffness, and altered hydration and charge dynamics. These phenomena also involve the disruption of collagen lamellar arrangement, allowing cancer cells and EVs to infiltrate the surrounding tissue. Radial collagen fiber structures observed in MIBC are derived from the deformation of circular collagen fibers, facilitating collective cancer cell invasion and immune cell infiltration. These changes give rise to TACS that transition from barrier-forming (TACS-II) in DCIS to invasion-guiding (TACS-III) structures during MIBC. Collectively, these ECM alterations facilitate cancer cell invasion, modulate immune infiltration, and contribute to the formation of a pre-metastatic niche. These findings suggest that structural changes in type I collagen and the ECM, driven by interactions among cancer, stromal, and immune cells, play a critical role in facilitating tumor progression, thereby potentially guiding future therapeutic strategies to target the tumor microenvironment.

## Figures and Tables

**Figure 1 cells-14-01037-f001:**
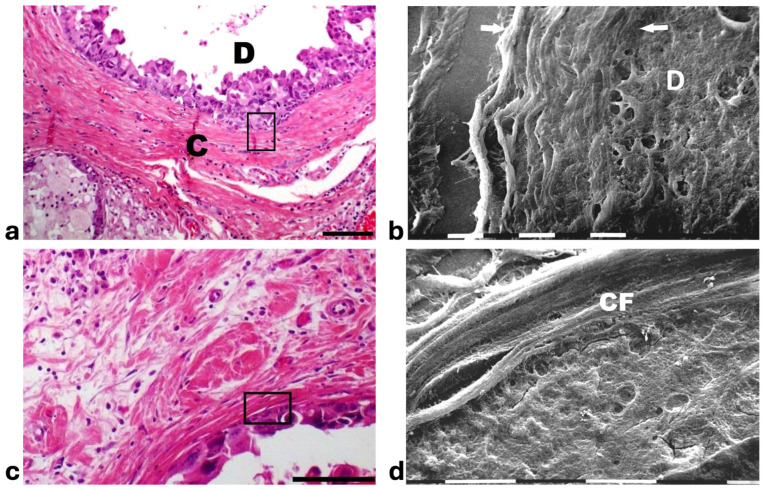
**Correlative SEM analysis of deparaffinized histological sections of DCIS and TACS-II stage human biopsies.** (**a**) The hematoxylin-eosin histological section shows a transversally sectioned duct (**D**) including proliferating cancer cells which do not invade the ECM, as circular densely packed collagen fibers (**C**) running parallel to the BM which are able to oppose cancer cells migration. No large inter-fiber spaces are detectable very next to the duct and only a poor immune infiltrate is visible only far from the duct. Bar = 50 µm. (**b**) A particular region delimited by the rectangular box in the previous deparaffinized histological section analyzed by SEM shows proliferating polygonal breast cancer cells inside a duct (**D**), surrounded by circular collagen fibers (between the two arrows) including densely packed fibrils and running parallel to the duct surface. No gaps or nano spaces are detectable between the ductal cancer cells and collagen fibers. White bar = 10 µm. (**c**) Breast cancer human biopsy of DCIS corresponding to TACS-II. A small portion of a duct containing cancer cells (on the bottom right side of the picture) is surrounded by densely packed circular collagen fibers running parallel along the duct surface. Among the circular collagen fibers, very small fusiform gaps are visible. Bar = 50 µm. (**d**) A particular area delimited by the rectangular box in the previous deparaffinized histological section analized by SEM. Many breast cancer cells inside a duct (on the below right side) are confined by a circular collagen fiber composed of densely packed collagen fibrils (**CFs**) which strongly adhere to the cancer cell BM, but a small fusiform inter-fibrillar space 5X-12 µm wide (on the left) is detectable next to the duct. White bar = 10 µm.

**Figure 2 cells-14-01037-f002:**
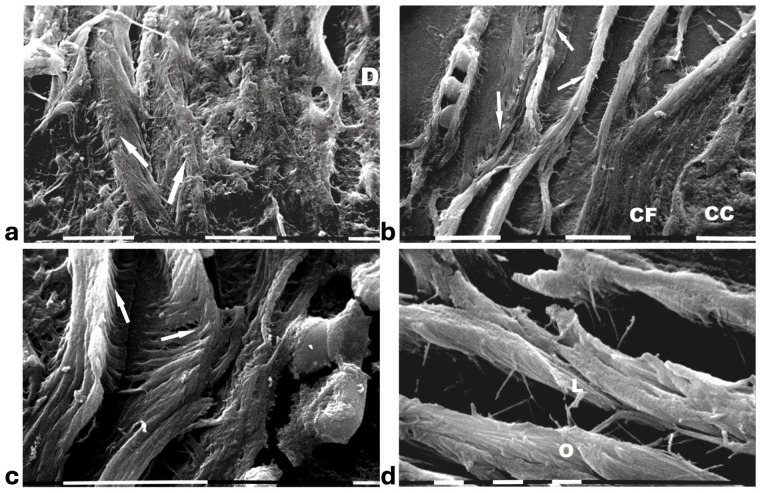
**SEM analysis of deparaffinized histological sections of DCIS and TACS II human biopsies.** (**a**) The collagen lamellae adjacent to the duct surface (**D**) appear densely packed and contain bundled fibrils running in the same direction and parallel inside each lamella, but orthogonal vs. adjacent lamellae (arrows). White bar = 10 µm. (**b**) The circular collagen fibers (**CFs**) described in the literature as fibers surrounding and successfully confining breast cancer cells (**CCs**) correspond to circular concentric collagen lamellae containing fibrils showing an orthogonal array vs. adjacent lamellae. In this picture the single lamellae are separated by less than 10 µm-wide inter-fiber spaces. A small vessel containing three immune cells is visible on the left side of the picture. White bar = 10 µm. (**c**) At higher magnifications the orthogonal array of the fibrils in adjacent collagen lamellae (arrows) is evident. A duct is visible on the right side. White bar = 10 µm. (**d**) The circular or concentric lamellae surrounding a duct appear composed of collagen fibers separated by inter-lamellar spaces and include densely packed fibrils running longitudinally (**L**) or obliquely (**O**) vs. the duct axis. White bar = 1 µm.

**Figure 3 cells-14-01037-f003:**
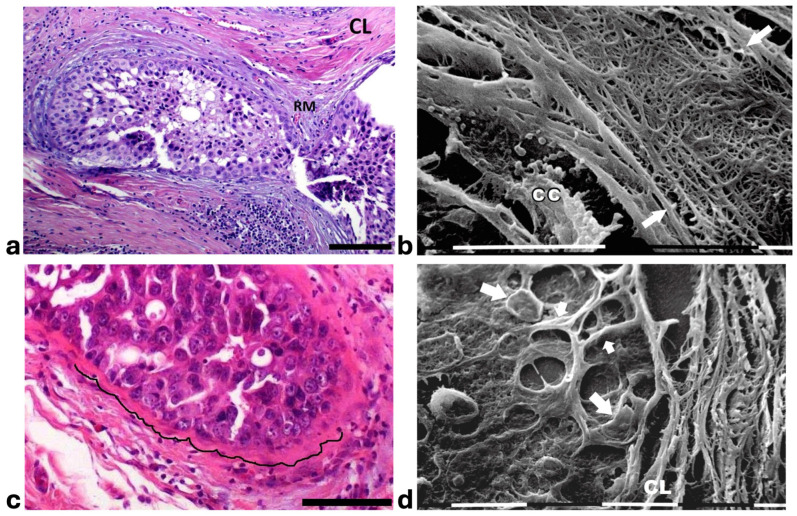
**Correlative microscopy by SEM analysis of deparaffinized histological sections of a DCIS and TACS-II stage human biopsy.** Human biopsy corresponding to DCIS and TACS-II stage. (**a**) In the hematoxylin-eosin histological section, two adjacent ducts, completely filled with proliferating breast cancer cells, seem to fuse and appear surrounded by densely packed circular collagen lamellae (**CL**) and peri-tumoral desmoplastic reticular collagen meshwork (**RM**). No cancer cell invading the ECM is detectable. Bar = 50 µm. (**b**) A particular of the previous histological section, deparaffinized and analyzed by SEM, shows a portion of a duct containing a breast cancer cell (**CC**) shedding microvesicles which are confined by densely packed fibrils. Both the cancer cell and microvesicles are confined by densely packed circular fibrils which, adhering to the BM, seem to progressively substitute a thin (about 10 µm) layer of a fibril meshwork (between the two arrows) of large fibrils (100–120 nm) probably corresponding to initial TACS-I stage or desmoplasia. White bar = 10 µm. (**c**) The histological section shows a duct filled with densely packed proliferating cancer cells. The jamming ductal cancer cells protrude towards the ECM so that the epithelial basal line, surrounded by circular collagen lamellae, appears micro-mamillated. The black line delineates the ductal contour, following a wavy trajectory. Bar = 50 µm. (**d**) Small area of the (**c**) histological section, deparaffinized and analyzed by SEM. Ductal cancer cells show a rounded concave shape (large arrows) and the rounded empty spaces might correspond to detached or deeper cancer cells. Collagen fibers (small arrows) developing from the circular concentric **CL** penetrate the epithelial basal layer and surround the peripheral ductal cancer cells. Fusiform spaces are visible among the collagen fibrils. White bar = 10 µm.

**Figure 4 cells-14-01037-f004:**
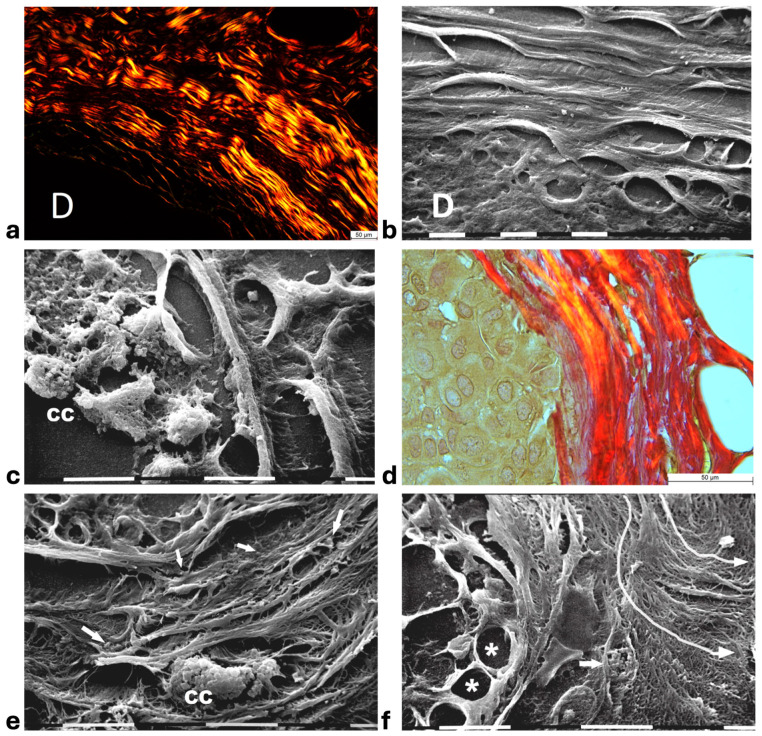
**Correlative microscopy by SEM analysis of deparaffinized histological sections of DCIS and TACS-II stage human biopsies.** Human biopsy of DCIS and a TACS-II stage. (**a**) The histological section stained with Picro Sirius Red to evidence collagen observed under the polarized light microscope shows a portion of a duct (**D**) surrounded by circular but crimped densely buckled collagen lamellae which show inter-lamellar spaces/channels. Bar = 50 µm. (**b**) Ultrastructural observation of a deparaffinized histological section evidences the densely packed concentric collagen lamellae surrounding a duct filled with proliferating cancer cells and showing inter-lamellar spaces/channels due to a partial inter-fiber dissociation. The fibril orthogonal array in adjacent collagen lamellae is still observable (**D**). Bar = 10 µm. (**c**) A deparaffinized histological section analyzed by SEM shows that when the concentric collagen lamellae (on the right) in contact with cancer cells are not interrupted, the proliferating cancer cells (**CCs**), also displaying many microvesicles on their surface, are successfully confined. Bar = 10 µm. (**d**) Histological section stained with Picro Sirius Red observed under the polarized light microscope to evidence collagen fibers. The proliferating polygonal jamming cancer cells confined in a duct try to invade the concentric collagen lamellae which show an early partial disruption and dissociation (in the center of the picture). Bar = 50 µm. (**e**) By SEM analysis the CL next to the BM of the duct are partially interrupted and show a discontinuity which allowed exosomes (small arrows) and microvesicles (large arrows) to pass through and invade the ECM. A cancer cell covered and surrounded by many microvesicles (**CC**) is visible among the circular and concentric collagen lamellae. Bar = 10 µm. (**f**) In some areas of the same histological section observed with SEM, the CL seem to bend and dramatically change their direction assuming a radial array (very long arrows) vs. the duct surface (on the left) and corresponding to a TACS-III stage. Collagen fibers forming thin sheaths surround the cancer cells of the basal layer (*). Extravesicles are detectable inside an inter-fiber channel (arrow). Both longitudinally (top part of the picture) and transversally cut fibrils (bottom part of the picture) of the dissociated circular concentric collagen CL are visible in the ECM. Bar = 10 µm.

**Figure 5 cells-14-01037-f005:**
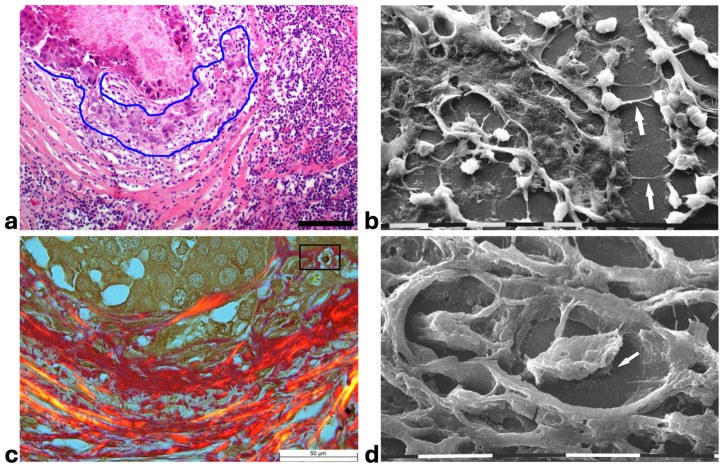
**Correlative microscopy by SEM investigation of deparaffinized histological sections of a MIBC and TACS-II stage human biopsy.** (**a**) In the hematoxylin–eosin histological section, the pressure of the jamming proliferating cancer cells completely filling the duct induces ECM invasion which favors an unjammimg state. A collective invasion of grouped cells is evident inside the area delimited by the blue line; the concentric peri-tumoral collagen lamellae show dissociation and inter-lamellar gaps in which both cancer cells and many immune cells are detectable. A conspicuous parvicellular infiltrate is present in inter-fiber spaces of the external collagen lamellae. Bar = 50 µm. (**b**) The previous histological section deparaffinized and evaluated by SEM shows that a collective ductal cancer cell invasion occurs where concentric collagen lamellae are separated by gaps filled with many rounded-shaped immune cells. Cytoplasmic processes similar to tunneling nanotubes (arrows) are evident among the adjacent rounded-shaped immune cells (presumably TAMs; upper arrow), as well as between immune and invading cancer cells (lower arrow). Bar = 10 µm. (**c**) Histological section stained with Picro Sirius Red to evidence collagen at the polarized light microscope. The jamming cancer cells inside the duct are invading the peri-tumoral ECM and show direct contact with the dissociated concentric collagen lamellae. Bar = 50 µm. (**d**) A particular area delimited by the rectangular box in the previous histological section deparaffinized and analyzed by SEM shows an invading ductal rounded cancer cell surrounded by and in tight contact with the dissociated collagen lamellae. The cancer cell develops short and ventral cytoplasmic processes similar to invadopodia (arrow). Bar = 10 µm.

**Figure 6 cells-14-01037-f006:**
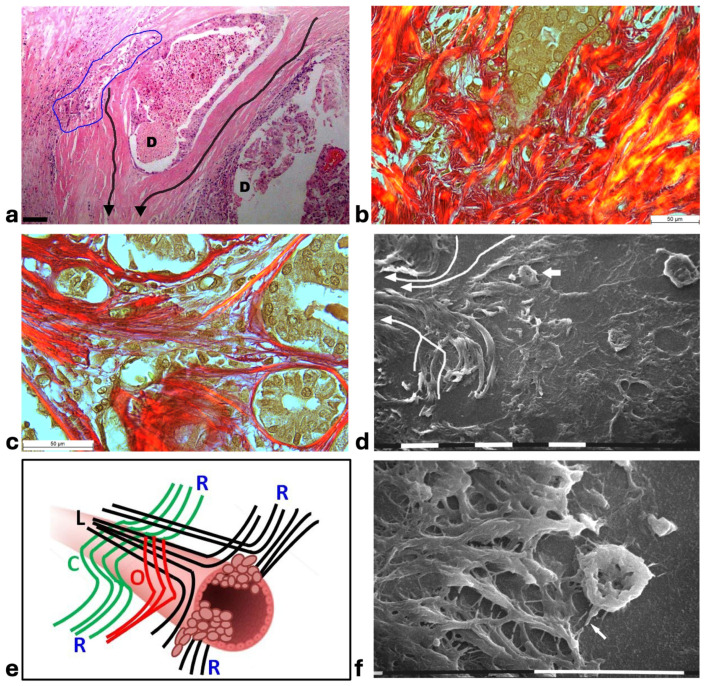
Correlative microscopy by SEM analysis of deparaffinized histological sections of a MIBC and TACS-III stage human biopsy. (**a**) The histological section shows two mammary gland ducts (**D**) which are filled with proliferating cancer cells. Around the duct at the left side, densely packed and concentric collagen lamellae appear in continuity with straight and radially aligned collagen lamellae (very long arrows). Cancer cell invasion occurs in the gaps of dissociated concentric collagen lamellae, inside the blue line. An immune cell infiltrate is clearly observable around the invading cancer cells. Bar= 50 µm. (**b**) The same sample with Picro Sirius Red staining observed under the polarized light microscope shows grouped ductal cancer cells penetrating into the deep peri-tumoral ECM by crossing disrupted and randomly arranged collagen lamellae in red. Bar= 50 µm. (**c**) In a Picro Sirius Red stained section observed under the polarized light microscope, the concentric collagen lamellae surrounding small ducts are in continuity with straight radially aligned collagen fibers which form a wide conic track or channel for cancer cell invasion. Bar = 10 µm. (**d**) By SEM analysis the deparaffinized previous section demonstrates that the concentric collagen lamellae parallel to the surface of a duct (the right-hand side) change their direction, becoming radially aligned collagen lamellae (very long arrows). A cancer cell (small arrow) seems to penetrate along the inter-fiber channel. Bar = 10 µm. (**e**) The radially aligned collagen fibers (**R**) described in TACS-III stage could derive from the dissociation of the concentric collagen lamellae, which include circular (**C**), oblique (**O**), and longitudinal (**L**) fibers simply changing their direction. (**f**) A concave and rounded-shaped cancer cell develops invadopodia (arrow) and filopodia, which attach to the straight collagen fibrils of swollen fibers (on the left side) and thus begin to invade along narrow inter-fiber channels. Bar = 10 µm.

**Figure 7 cells-14-01037-f007:**
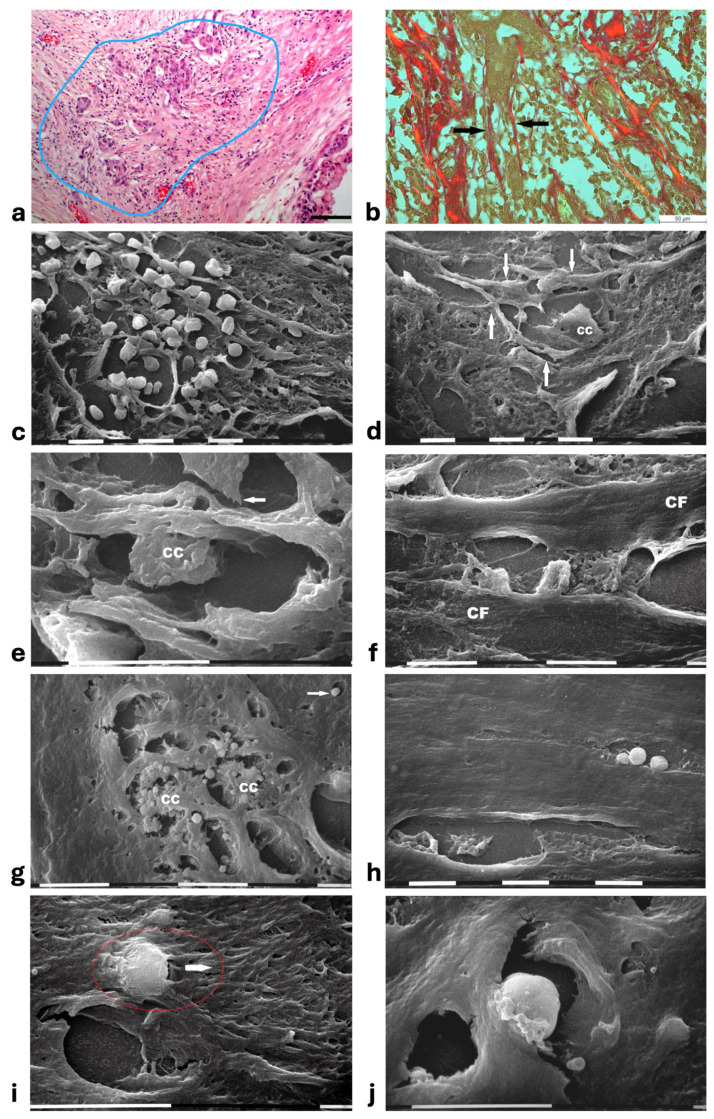
**Correlative microscopy by SEM analysis of deparaffinized histological sections of a MIBC and TACS-III stage human biopsy.** (**a**) The hematoxylin-eosin-stained histological section includes a portion of a duct (on the right) surrounded by CL which are not invaded by the cancer cells. However, in the outer layer of peri-tumoral ECM, dense bundles of collagen fibers following different directions are invaded by clusters of cancer cells (area inside the blue line), which seem to look for the straight outer radial fibers (on the left). Bar = 50 µm. (**b**) Histological section stained with Sirius Red and observed under the Polarized Light Microscope. Thin, straight, and aligned parallel collagen fibers which appear stained in red (arrows) delimit a cylindric channel for both cancer cell invasion (from the upside duct) and immune cell migration. Bar = 50 µm. (**c**) The deparaffinized section analyzed by SEM shows a massive infiltrate of immune cells floating among the straight and parallel collagen fibers. White bar = 10 µm. (**d**) In the same deparaffinized section observed with SEM, a polygonal invading cancer cell (**CC**) is penetrating between parallel, thin, and straight radially arranged fibers (arrows). The cancer cell strongly adheres to collagen fibers but also looks for adhesion to collagen by developing cytoplasmic protrusions (on its left side). Bar = 10 µm. (**e**) Two polygonal single cancer cells invading the peri-tumoral ECM intimately adhere to the collagen lamellae composed of collagen fibers, show extravesicles on their cytoplasmic surface, and develop invadopodia (arrow). White bar = 10 µm. (**f**) By SEM analysis two single and thin collagen fibers (**CFs**) including very densely packed collagen fibrils delimit a long inter-fiber channel (about 10 µm wide) containing polygonal-shaped cancer cells which produce extravesicles, display cell–cell contact and stick to the collagen wall of the channel. White bar = 10 µm. (**g**) By SEM analysis an inter-fiber channel transversally cut contains two cancer cells (**CCs**) producing and releasing extravesicles. Note a single microvesicle in a relative space inside the collagen fibers (arrow). White bar = 10 µm. (**h**) A deparaffinized section observed by SEM shows that very dense collagen fibers of ECM, far from the ducts, do not show the single fibrils they are composed of, and confine inter-fiber channels which contain cancer cells (downside) and three immune cells (upside). White bar = 10 µm. (**i**) By SEM analysis a single rounded-shaped immune cell almost completely encapsulated and covered by dense collagen fibers (in the red circle) is degrading and invading the peri-tumoral collagen ECM, following the arrow direction. An empty hole or inter-fiber channel in ECM is visible below. Black Bar = 10 µm. (**j**) In a deparaffinized section analyzed by SEM, a single immune cell strongly adhering to the wall of an inter-fiber channel develops cytoplasmic protrusions to migrate into a preformed ECM channel. Far from the cell, a few isolated microvesicles are also visible. White bar = 10 µm.

**Figure 8 cells-14-01037-f008:**
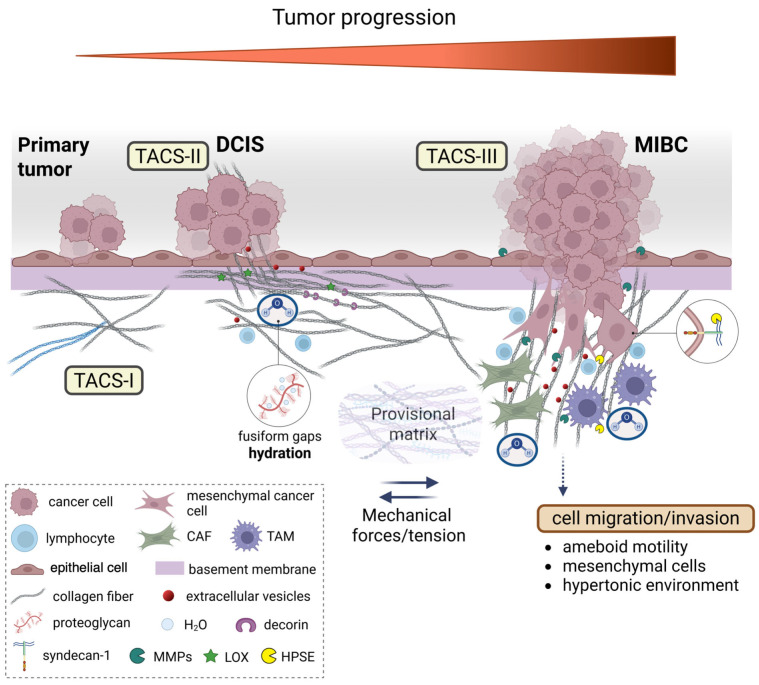
**Dynamic changes in collagen fibril organization during breast cancer progression.** The transition from ductal carcinoma in situ (DCIS) to microinvasive breast carcinoma (MIBC) involves the profound remodeling of the tumor microenvironment (TME), particularly the peri-tumoral collagen architecture. In the early stages, epithelial cells of the primary tumor are surrounded by concentric collagen lamellae composed of densely packed fibrils arranged orthogonally across adjacent layers. This organization generates a biophysical barrier that resists tumor expansion via tension forces and structural compaction. Tumor-derived paracrine signals trigger a desmoplastic reaction, leading to the activation of resident fibroblasts into cancer-associated fibroblasts (CAFs), which promote collagen cross-linking mainly through the lysyl oxidase (LOX) function, increase matrix stiffness, and alter local charge distributions through enzymatic remodeling by matrix proteolytic enzymes, such as metalloproteinases (MMPs). Concurrently, tumor-associated macrophages (TAMs) infiltrate the stroma and contribute to collagen degradation and reorganization. These immune and stromal interactions reshape the hydration state of the extracellular matrix (ECM), as proteoglycans (PGs) and glycosaminoglycans (GAGs), mainly hyaluronan (HA), modulate water content, viscoelasticity, and osmotic pressure that influence both cell motility and matrix mechanosensing. As invasive cancer cells invade the basement membrane (BM), they encounter a mechanically and chemically altered stroma facilitating the alteration in the provisional matrix and the formation of a pre-metastatic niche. Collagen fibers adjacent to these cells display tumor-associated collagen signatures (TACS). In TACS-II, collagen fibers align parallel to the tumor border, creating physical barriers; in TACS-III, they become linearized and oriented perpendicularly, forming channels that guide cancer cell migration/invasion and the initiation of metastasis. These structural changes are influenced by local tension forces, changes in charge density of matrix macromolecules, and hydration-dependent phenomena of interfibrillar spaces. These microenvironmental alterations not only facilitate cancer cell invasion but also shape immune cell infiltration and extracellular vesicle (EV) secretion, thereby reinforcing breast cancer progression.

**Table 1 cells-14-01037-t001:** Clinical samples used for histological and scanning electron microscopy (SEM) analysis.

Case No.	Age	Diagnosis	Pathological Stage [52]	Explanation
18-12254A3	36	High Grade DCIS	pTis	High Grade DCIS
18-12254A8	40	High Grade DCIS	pTis	High Grade DCIS
20-968Z2	68	High Grade DCIS	pTis	High Grade DCIS
19-6424Γ2	67	MIBC	pT1mi	Tumor ≤ 1 mm in greatest dimension
12-560Γ6	60	MIBC	pT1mi	Tumor ≤ 1 mm in greatest dimension
12-560Γ7	59	MIBC	pT1mi	Tumor ≤ 1 mm in greatest dimension

## Data Availability

The raw data supporting the conclusions of this article will be made available by the corresponding authors on request.

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
