# Peer review of "Ultrastructural Changes of the Peri-Tumoral Collagen Fibers and Fibrils Array in Different Stages of Mammary Cancer Progression"

_cells, 2025, doi:10.3390/cells14131037_

Round 1
Reviewer 1 Report
Comments and Suggestions for Authors
In this study, Marco Franchi et al. investigated the role of peritumoral collagen architecture in breast cancer progression using correlative scanning electron microscopy (SEM) of human specimens. In ductal carcinoma in situ (DCIS) they identify concentric lamellae of type I collagen fibrils parallel to the ducts and orthogonal orientation between adjacent layers. These structures associated with absence of cancer cell invasion. On the contrary, minimally invasive breast carcinoma (MIBC) were characterized by lamellar dissociation and interfiber gaps likely responsible of cancer cell and exosome infiltration, radial reoorganization of fibers facilitating invasion and immune cell recruitment. The author propose that the crosstalk between stroma cancer and immune cells drives ECM remodeling positioning peritumoral collagen architecture as a central mediator of early invasion.
Comments
This manuscript presents a compelling study that underscores the critical role of collagen rearrangements in metastatic progression. The authors support their findings with thorough and detailed analyses and, the included images are not only striking but also highly effective in visually reinforcing the key conclusions of the work. However I consider that:
A: Although the tumor samples are limited in number, providing a quantitative analysis of the observed morphological features and tumor/immune cells, would significantly strengthen the study. Also the delimitations of the area of interest within the figures (Fig 5-,6-,7-A, 7-i) could be improved.
B: The manuscript requires comprehensive language editing to meet publication standards. To assist with this, the authors may consider using AI-based writing tools (some of which are freely available) for initial improvements in grammar, clarity, and style. While I highlight representative examples below, these issues are pervasive throughout the manuscript (particularly the abstract, introduction and methods), and warrant systematic review:
-Line 17; I would speak about changes that are associated to cancer progression since there is no demonstration of an actual contribution in this study.
-Lines 30-31; Consider: “Understanding the mechanisms that drive the progression from…of research, crucial for understanding tumor invasion and metastatic potential”
-Lines 33-35; consider: To investigate the progression from DCIS to MIBC, we analyzed peritumoral collagen architecture using correlative scanning electron microscopy (SEM) on histological sections from human biopsies”. In general for scientific precision I would recommend using 'collagen architecture' rather than 'collagen array' throughout this manuscript.
-Lines 35-39; Consider: In DCIS, the peritumoral collagen organizes into concentric lamellae ('circular fibers') parallel to the ducts. Within each lamella, type I collagen fibrils align in parallel, while neighboring lamellae show orthogonal fiber orientation. The concentric lamellar arrangement of collagen may physically constrain cancer cell migration, explaining the lack of visible tumor cell invasion into the peritumoral ECM in DCIS.
lines 68-70; Consider: The fibrils exhibit hierarchical organization with alternating chirality at both molecular and supramolecular levels. This rope-like structural configuration provides enhanced resistance to tensile forces.
-Lines 73-76; Consider: Notably, collagen architecture directly regulates cell migration: densely bundled fibrils promote stable fibroblast adhesion, while pliable reticular fibrils facilitate adhesion retraction. This structural dichotomy suggests that extracellular matrix organization serves as a biomechanical switch governing migratory behavior. Moreover, Increased ECM stiffness represents a prognostically adverse microenvironment that promotes cancer cell invasion and metastasis.
_Lines 76-78; Rewrite and avoid repetitions.
-Lines 80-83; Consider: Breast cancer cells synthesize an aberrant homotrimeric type I collagen that assembles into thinner yet mechanically stiffer fibrils. These tumor-derived collagen structures exhibit dual pathological properties: (1) they actively promote cancer cell migration, and (2) are resistant to matrix metalloproteinase (MMP) degradation, a feature conserved across multiple solid tumor types.
-Lines 86-88¸Consider: Tumor progression requires extensive remodeling of the peritumoral collagen architecture, which can biomechanically inhibit or promote cancer cell invasion.
-Line 395; Fig. 7i;j
Line 432; Consider: “Collagen lamellae are..”
Line 561; Consider: “empty interfiber spaces..”
Overall, this is a valuable study, but refinements in sentence structure, conciseness, and grammar would further enhance its clarity and impact.
Comments on the Quality of English LanguageQuality of English language should be improved
Author Response
Reviewer 1
We would like to express our gratitude in this reviewer for the good words for our experimental study and for providing a positive and constructive criticism. All indicated points by the reviewer have been taken into consideration as to improve the clarity and the quality of the manuscript.
Point-by-point responses to Reviewer’s comments:
A: Although the tumor samples are limited in number, providing a quantitative analysis of the observed morphological features and tumor/immune cells, would significantly strengthen the study. Also the delimitations of the area of interest within the figures (Fig 5-,6-,7-A, 7-i) could be improved.
We thank the reviewer for this important point. Actually, in this study we had the possibility to process human biopsies performer for histological diagnosis, selecting similar breast cancer types, with similar diagnosis, or common anatomo-pathological aspects, which is not an easy approach indeed. The limited number of the samples (3+3), the very different age patients (36-69) and the individual immuno response do not suggest to elaborate a statistical-quantitative analysis of the data. Therefore, we presented a very carefull qualitative morphological analysis with results which even with some individual differences, were the same inside the two groups (DCIS and MIBC). The idea behind this paper is to introduce the ultrastructural aspects of the nano-microenvironment peri-tumoral collagen architecture and their biomechamical/biochemical roles as critical factors in regulating cancer cell invasion. The deliminations in the proposed figures have been improved, accordingly.
B: The manuscript requires comprehensive language editing to meet publication standards. To assist with this, the authors may consider using AI-based writing tools (some of which are freely available) for initial improvements in grammar, clarity, and style. While I highlight representative examples below, these issues are pervasive throughout the manuscript (particularly the abstract, introduction and methods), and warrant systematic review:
We thank the Reviewer 1 and we provided to an english language editing review.
-Line 17; I would speak about changes that are associated to cancer progression since there is no demonstration of an actual contribution in this study.
Thanks for this comment too: We have now modified accordingly the first sentence in Abstract: “Breast cancer invasion and subsequent metastasis to distant tissues occur when cancer cells loose cell-cell contact, develop a migrating phenotype, and invade the basement membrane (BM) and the extracellular matrix (ECM) to penetrate blood and lymphatic vessels”.
-Lines 30-31; Consider: “Understanding the mechanisms that drive the progression from…of research, crucial for understanding tumor invasion and metastatic potential”
We changed the sentence in line 44-47 as suggested: “Understanding the mechanisms that drive the progression from DCIS to MIBC are crucial to understand tumor invasion and prevent metastasis [3]”.
-Lines 33-35; consider: To investigate the progression from DCIS to MIBC, we analyzed peritumoral collagen architecture using correlative scanning electron microscopy (SEM) on histological sections from human biopsies”. In general for scientific precision I would recommend using 'collagen architecture' rather than 'collagen array' throughout this manuscript.
We substituted the sentence in line 22-24 as the Reviewer asked: “To investigate the progression from DCIS to MIBC, we analyzed peritumoral collagen architecture using correlative scanning electron microscopy (SEM) on histological sections from human biopsies”, accordingly.
-Lines 35-39; Consider: In DCIS, the peritumoral collagen organizes into concentric lamellae ('circular fibers') parallel to the ducts. Within each lamella, type I collagen fibrils align in parallel, while neighboring lamellae show orthogonal fiber orientation. The concentric lamellar arrangement of collagen may physically constrain cancer cell migration, explaining the lack of visible tumor cell invasion into the peritumoral ECM in DCIS.
We modified according to the reviewer’s suggestion the sentence in line 24-28:” “In DCIS, the peritumoral collagen organizes into concentric lamellae ('circular fibers') parallel to the ducts. Within each lamella, type I collagen fibrils align in parallel, while neighboring lamellae show orthogonal fiber orientation. The concentric lamellar arrangement of collagen may physically constrain cancer cell migration, explaining the lack of visible tumor cell invasion into the peritumoral ECM in DCIS.”
lines 68-70; Consider: The fibrils exhibit hierarchical organization with alternating chirality at both molecular and supramolecular levels. This rope-like structural configuration provides enhanced resistance to tensile forces.
Thank you for this comment too. We have changed accordingly the sentence in line 57-59 as suggested: “The fibrils exhibit hierarchical organization with alternating chirality at both molecular and supramolecular levels. This rope-like structural configuration provides enhanced resistance to tensile forces. [27–30]”.
-Lines 73-76; Consider: Notably, collagen architecture directly regulates cell migration: densely bundled fibrils promote stable fibroblast adhesion, while pliable reticular fibrils facilitate adhesion retraction. This structural dichotomy suggests that extracellular matrix organization serves as a biomechanical switch governing migratory behavior. Moreover, Increased ECM stiffness represents a prognostically adverse microenvironment that promotes cancer cell invasion and metastasis.
We developed the sentence in line 62-65 as suggested: “Notably, collagen architecture directly regulates cell migration: densely bundled fibrils promote stable fibroblast adhesion, while pliable reticular fibrils facilitate adhesion retraction [33].This structural dichotomy suggests that extracellular matrix organization serves as a biomechanical switch governing migratory behavior. Moreover, increased ECM stiffness represents a prognostically adverse microenvironment that promotes cancer cell invasion and metastasis”.
-Lines 76-78; Rewrite and avoid repetitions
It has been corrected accordingly. Particularly, in line 65-68 we wrote the new sentence to avoid repetitions:” A denser tumor microenvironment including stiff fibrils of type I collagen promotes the growth of cytoplasmic protrusions and high-traction forces ability in cancer cells which assume migrating phenotypes [8–12,34–36].”
-Lines 80-83; Consider: Breast cancer cells synthesize an aberrant homotrimeric type I collagen that assembles into thinner yet mechanically stiffer fibrils. These tumor-derived collagen structures exhibit dual pathological properties: (1) they actively promote cancer cell migration, and (2) are resistant to matrix metalloproteinase (MMP) degradation, a feature conserved across multiple solid tumor types.
We changed the sentence in line 69-72 as suggested: “Breast cancer cells synthesize an aberrant homotrimeric type I collagen that assembles into thinner yet mechanically stiffer fibrils. These tumor-derived collagen structures exhibit dual pathological properties: (1) they actively promote cancer cell migration, and (2) are resistant to matrix metalloproteinase (MMP) degradation, a feature conserved across multiple solid tumor types.[37–40].”
-Lines 86-88¸Consider: Tumor progression requires extensive remodeling of the peritumoral collagen architecture, which can biomechanically inhibit or promote cancer cell invasion.
We modified the sentence in line 75-77 as suggested: “Tumor progression requires extensive remodeling of the peritumoral collagen architecture, which can biomechanically inhibit or promote cancer cell invasion [42,43].”
-Line 395; Fig. 7i;j
We corrected the mistake in line 384.
-Line 432; Consider: “Collagen lamellae are..”
We corrected the grammar mistake in line 421.
-Line 561; Consider: “empty interfiber spaces..”
We corrected the mistake in line 550.
Overall, this is a valuable study, but refinements in sentence structure, conciseness, and grammar would further enhance its clarity and impact.
We have made several refinement points indicated as to improve the manuscripts clarity and quality, accordingly.
Reviewer 2 Report
Comments and Suggestions for Authors
The authors of the manuscript titled “Ultrastructural changes of the peri-tumoral collagen fibers and 2 fibrils array in different stages of mammary cancer progression” attempts to correlate the collagen fiber alignment with the progression of mammary cancer using SEM. The results shown in this manuscript directly visualize the connection between TACS and cancer cell metastasis, which is worth documenting at least. I only have two minor points listed below:
- The introduction is a solid, comprehensive review that accurately covers the role of aligned collagen in cancer progression. One publication about cancer cell response to aligned collagen fibers should be cited: “Directional cues in the tumor microenvironment due to cell contraction against aligned collagen fibers”, PMID 33965625
- Figure 7 shows a striking image of what appears to be infiltrating immune cells within collagen bundles. That being said, it’s not clear how they identified the cell type—there’s no mention of immunofluorescence markers or other validation. Without markers (e.g., CD3 for T cells, CD68 for macrophages), it’s hard to know whether these are lymphocytes, macrophages, fibroblasts, or even cancer cells. It would be better to include some IF/marker data to support the identity of these cells. If not, the authors should clarify the ambiguity here.
Author Response
Reviewer 2:
We would like to thank this reviewer for the good words and for providing a positive feedback. All indicated points have been taken into consideration as to improve the quality of the manuscript.
Point-by-point responses to Reviewer’s comments:
The authors of the manuscript titled “Ultrastructural changes of the peri-tumoral collagen fibers and 2 fibrils array in different stages of mammary cancer progression” attempts to correlate the collagen fiber alignment with the progression of mammary cancer using SEM. The results shown in this manuscript directly visualize the connection between TACS and cancer cell metastasis, which is worth documenting at least. I only have two minor points listed below:
- The introduction is a solid, comprehensive review that accurately covers the role of aligned collagen in cancer progression. One publication about cancer cell response to aligned collagen fibers should be cited: “Directional cues in the tumor microenvironment due to cell contraction against aligned collagen fibers”, PMID 33965625
We included the Reference suggested by the Reviewer in the Discussion: “Szulczewski JM, Inman DR, Proestaki M, Notbohm J, Burkel BM & Ponik SM. (2021) Directional cues in the tumor microenvironment due to cell contraction against aligned collagen fibers. Acta Biomater. 129, 96-109.”
- Figure 7 shows a striking image of what appears to be infiltrating immune cells within collagen bundles. That being said, it’s not clear how they identified the cell type—there’s no mention of immunofluorescence markers or other validation. Without markers (e.g., CD3 for T cells, CD68 for macrophages), it’s hard to know whether these are lymphocytes, macrophages, fibroblasts, or even cancer cells. It would be better to include some IF/marker data to support the identity of these cells. If not, the authors should clarify the ambiguity here.
We thank the Reviewer for appreciating the Figure 7. The morphological phenotype allowed us to distinguish lymphocytes, macrophages, fibroblasts both at light microscope but much better at SEM where a higher magnification and quality is feasible. To clarify the point, fibroblasts show a mesenchymal phenotype with an elongated fusiform shape whereas the immuno cells display a globular shape. Lymphocytes have a small diameter of 6-10 µm vs. macrophage which show a larger diameter (21 µm). Other studies report volume differences: macrophages are able to develop tunneling nanotubes with cancer cells (as we have observed in the paper) and have a larger volume (5000 µm3) vs. lymphocytes (130 µm3).